## META-RESEARCH

# Incidences of problematic cell lines are lower in papers that use RRIDs to identify cell lines

**Abstract** The use of misidentified and contaminated cell lines continues to be a problem in biomedical research. Research Resource Identifiers (RRIDs) should reduce the prevalence of misidentified and contaminated cell lines in the literature by alerting researchers to cell lines that are on the list of problematic cell lines, which is maintained by the International Cell Line Authentication Committee (ICLAC) and the Cellosaurus database. To test this assertion, we text-mined the methods sections of about two million papers in PubMed Central, identifying 305,161 unique cell-line names in 150,459 articles. We estimate that 8.6% of these cell lines were on the list of problematic cell lines, whereas only 3.3% of the cell lines in the 634 papers that included RRIDs were on the problematic list. This suggests that the use of RRIDs is associated with a lower reported use of problematic cell lines.
DOI: https://doi.org/10.7554/eLife.41676.001

**ZELJANA BABIC, AMANDA CAPES-DAVIS, MARYANN E MARTONE, AMOS BAIROCH, I BURAK OZYURT, THOMAS H GILLESPIE AND ANITA E BANDROWSKI\***

## Introduction

Cell lines are widely used in the biological sciences, partly because they are able to multiply indefinitely. This property means that scientists can, in theory, exactly replicate previous studies and then build on these results (*American Type Culture Collection Standards Development Organization Workgroup ASN-0002, 2010*). However, mislabeling or mishandling can result in misidentification, contamination or distribution of problematic cell lines which, in turn, can affect the validity of research data. Despite this, testing for contamination and misidentification is commonly not performed and, therefore, a laboratory could obtain contaminated cell lines and spend significant resources pursuing research based on faulty premises (*Drexler et al., 2017*). A particular problem is the contamination of a cell line by a cancer cell line such as HeLa (*Capes-Davis et al., 2010*) because cancer cells tend to grow more rapidly than normal cells.

In 2012, a group of scientists established an organization called the International Cell Line Authentication Committee (ICLAC) to create a register of contaminated and misidentified cell lines, increase awareness of the problem, and propose approaches to decrease the use of such cell lines (*Masters, 2012*). ICLAC and other expert groups (such as the American Type Culture Collections: ATCC) propose that one of the most practical ways to detect contamination is to make use of a DNA-based technique called short-tandem-repeat profiling to authenticate human cell lines (*American Type Culture Collection Standards Development Organization Workgroup ASN-0002, 2010*). Additionally, the Cellosaurus database, housed at the Swiss Institute of Bioinformatics, contains extensive information on more than 100,000 cell lines (*Bairoch, 2018*), including information about cell line misidentification and other problems (such as incorrect tissue origin, which is not detected by short-tandem-repeat profiling).

Despite the availability of the ICLAC register, the overall rate of misidentified cell line use has not fallen across the literature (*Horbach and Halffman, 2017*). It has been shown that

**\*For correspondence:**
abandrowski@ncmir.ucsd.edu

publishers checking the manuscript during submission for misidentified cell lines can eliminate their use, although such a process requires significant time and cost (*Fusenig et al., 2017*). We believe that the continued use of misidentified cell lines in published studies is due, in part, to researchers not checking the lists of misidentified cell lines consistently.

Research Resource Identifiers (RRIDs) are unique identifiers that can be included in the methods section of a research paper to define the cell line, antibody, transgenic organism or software used (*Bandrowski et al., 2016*; *Bandrowski and Martone, 2016*). These identifiers were introduced largely because antibodies are a known source of variation in experiments, yet researchers did not include enough metadata to identify which antibody they used (*Vasilevsky et al., 2013*). RRIDs are slowly becoming more common in the literature, especially in some disciplines such as neuroscience. They are issued by the naming authority for a particular type of resource (for example, stock centers and model-organism databases issue RRIDs for organisms; Cellosaurus issues RRIDs for cell lines; the antibody registry issues RRIDs for antibodies; and the SciCrunch registry issues RRIDs for software and other digital resources).

RRIDs for cell lines were first incorporated into the RRID portal in 2016, and journals often require researchers to look up each antibody, animal or cell line on this website (which is synchronized with the Cellosaurus database, and therefore contains information about cell line contamination or misidentification). This requirement creates a natural experiment to address the question: if researchers are alerted about the misidentification of a cell line before they publish, will they still report data from such a cell line?

## Results

In this study, we used text mining to identify papers that included RRIDs and papers that listed cell lines, and then compared the prevalence of misidentified cell lines in these two samples. It should be stressed that the use of cell lines on the problematic list does not automatically mean that a given cell line is being employed improperly. For example, the problematic list includes cases where a cell line is 'partially contaminated', which does not affect cells purchased from, say, a stock center. The list

also includes cell lines that have been labeled with the wrong type of cancer, but these may still be safely used if the researchers know the true identity of the line. We must, therefore, exercise caution when interpreting these results. More information about the problematic list please is given in the discussion section and *Supplementary file 2*.

### Text mining corpus

To derive a dataset of reported cell lines from the general literature, we used text mining of the PubMed Central open access subset. This task requires the use of natural language processing, as cell-line names are not unique strings. For example, looking for a set of characters, such as 'H2', may reveal papers where H2 denotes a cell line, gene, protein, antibody or a figure. We used the SciScore tool, a 'Named Entity Recognition-based algorithm' specialized for scientific resources, which can in principle recognize the H2 cell line and can ignore its mentions referring to something other than a cell line.

The algorithm was taught to consider features around the recognized term that are associated with cell lines. SciScore was trained using 1,457 annotations made by a curator as well as a complete list of cell lines and synonyms from Cellosaurus as a set of seed data. We split the data into 90% training, and 10% testing of random sets ten times, and obtained an average precision of 87.3% (+/- sd 4.65%), recall 61.9% (+/-sd 7.01%) with an overall F1 mean of 72.2% (+/- sd 5.39%).

This algorithm was then deployed on the ~2 million articles in the open access subset of PubMed Central. It recognized 305,161 unique cell-line names from a total of 150,459 unique articles (*Figure 1*), which we treated as our population of cell lines that was considered for further analysis. These data were then provided to Cellosaurus curators, allowing them to create entries for 22 cell lines that had been so far overlooked, and to add 18 synonyms and eight misspellings for existing entries, making Cellosaurus a more complete resource.

As cell-line names are not standardized in most papers, we used three different approaches to estimate the percentage of problematic cell lines in the general literature. First, to obtain a lower boundary, we determined whether the cell line name, as stated by the

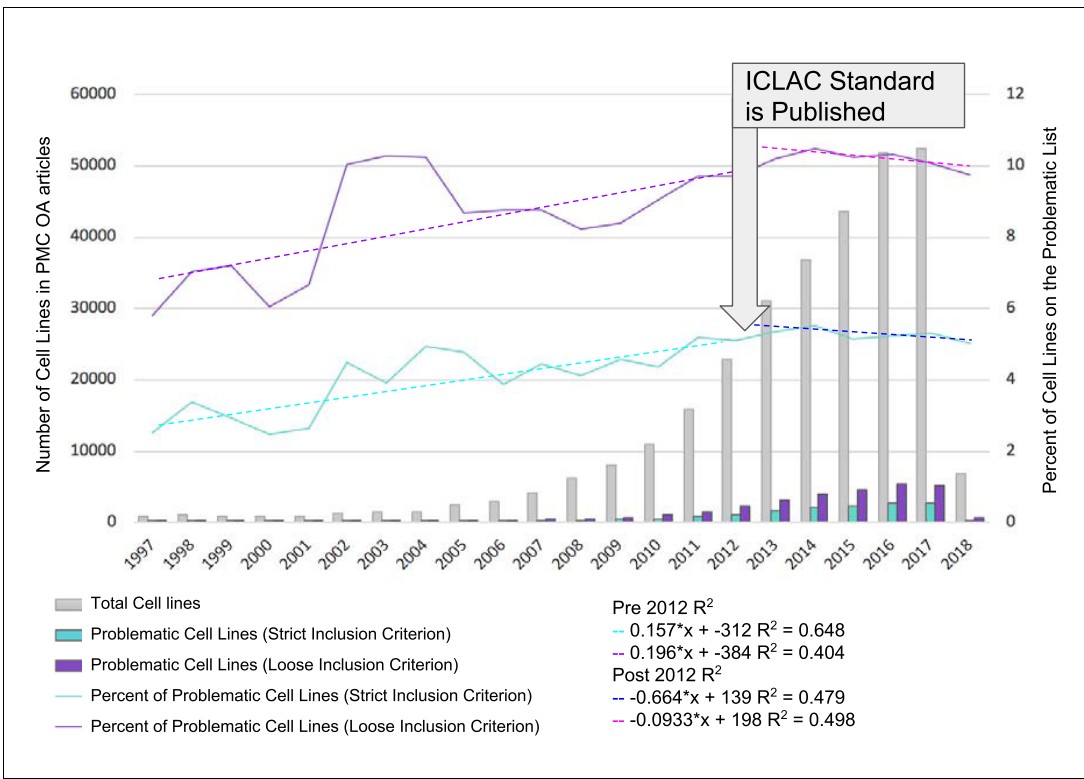

**Figure 1.** Identification of misidentified cell lines. The number of cell lines used in PubMed Central articles available for text mining is shown as a function of year. The names of cell lines were matched using two criteria, strict and loose. The strict criterion constitutes an exact match where the name used by the researchers and detected by SciScore is on the list of ICLAC register of misidentified cell lines. The loose criterion was calculated by adding a wild-card character (*) to the end of all names found by SciScore, and matching the names and synonyms on the ICLAC list. The graph is divided into two sections: before and after 2012. 2012 was chosen as the year to break the graph because the publication of the authentication standard and the formation of ICLAC occurred that year (**Masters, 2012**).

DOI: https://doi.org/10.7554/eLife.41676.002

The following source data is available for figure 1:

**Source data 1.** Underlying data for *Figure 1*.

DOI: https://doi.org/10.7554/eLife.41676.003

---

researcher, matched exactly a name or synonym of a problematic cell line in Cellosaurus, as this is the most stringent criterion (*Supplementary file 2*).

To determine an upper boundary, we used the cell line name and a wild-card character, so HeLa became HeLa*. In this loose condition, if researchers were to report "CF-1 mouse embryonic fibroblasts", which was recognized as the entity by SciScore, then the loose criterion would match this to CF-1. However, under the strict criterion, those would be considered different.

The edit distance was also employed to come up with another estimate intermediate between the two. In this condition, an edit distance of 0 was set when the letters and numbers of the cell line matched exactly, and the hyphens, spaces

and other special characters were omitted. If a researcher uses CF1 or CF-1, for example, the edit distance metric would consider these the same cell line. However, the "CF-10", "CAF-1" or "CF-1 mouse embryonic fibroblasts" would be an edit distance of more than 0, and therefore, would not match.

We first considered the reporting of problematic cell lines as a function of the publication year (*Figure 1*). We did not take into account data prior to 1997, since there were very few papers in the open access PubMed Central corpus that contained cell lines (<25 per year) (See *Figure 1—source data 1* for data including years prior to 1997). We split the data from 1997 to 2012 and 2013 to 2018, as the consensus recommendations for authentication of human cell

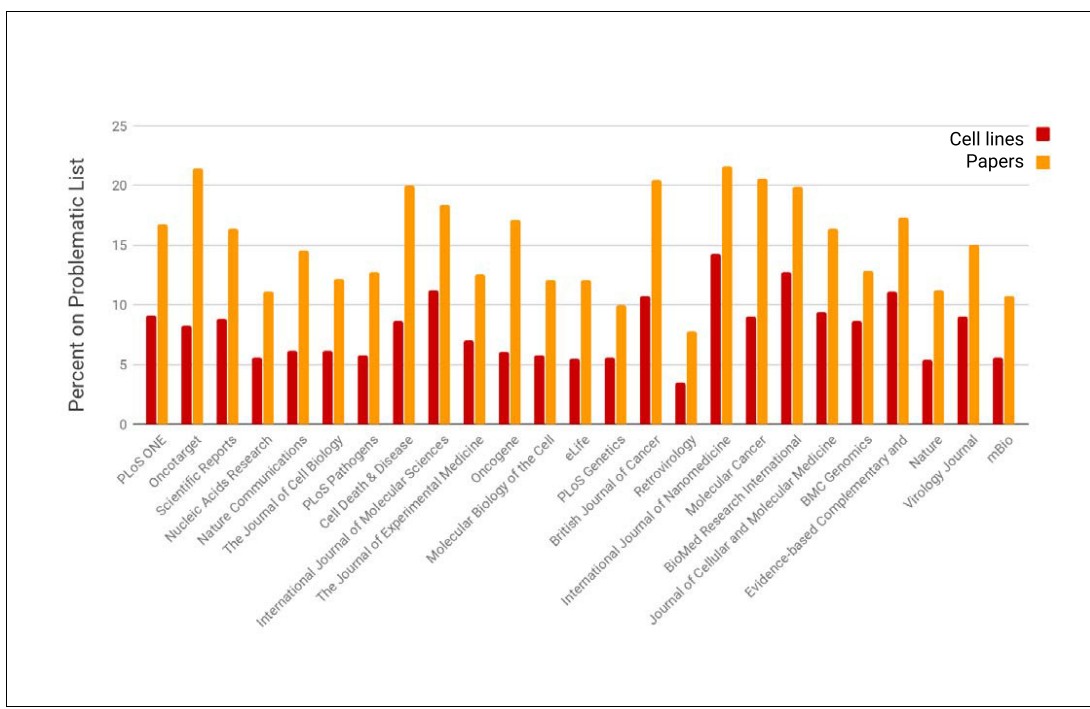

**Figure 2.** The prevalence of open access papers containing one or more cell lines found on the problematic list. Journals are sorted from left to right by the number of cell lines detected by SciScore (only the top 25 journals are shown for presentation purposes; data for all journals is given in *Figure 2—source data 1*). Each bar represents the percent of cell lines (red) or papers (orange) that are on the problematic cell-line list. Cell line presence on the misidentified list is scored by the edit distance metric, which skips all special characters such as spaces and dashes and assumes that any string that contains the same letters and numbers is an edit distance of 0 (e.g., EF 1 = EF-1). Journals that published papers under a license not allowing text mining are not represented here.
DOI: https://doi.org/10.7554/eLife.41676.004

The following source data is available for figure 2:

**Source data 1.** Data on problematic cell lines for all journals.
DOI: https://doi.org/10.7554/eLife.41676.005

lines were published in 2012 (*ATCC, 2011*; *Masters, 2012*).

Awareness of misidentified cell lines and authentication testing also received a boost with the establishment of the ICLAC in 2012. Ever since, the ICLAC has been providing a focus for change. Thus, although there is certainly a long history of reporting of cell line problems, 2012 can be viewed as the seminal breakpoint. Both the strict and loose criteria were considered as a function of year as the lower and upper bounds of contamination rates, and the $R^2$ calculation was used to assess whether there is a trend in the percentage of problematic cell lines used by year. From 1997 to 2012, the percent of reported cell lines on the problematic list appears to be increasing, although the $R^2$ shows a rather weak correlation (0.648 strict criterion or 0.404 loose criterion). In the period between 2013 and 2018, the slope is reversed, although still a rather weak correlation (0.479 strict and 0.498 loose), suggesting that there may indeed be a trend to decrease the use of problematic cell lines.

To assess the number of studies that used problematic cell lines, we asked what percentage of papers contain one or more cell lines where the name detected by SciScore matches a cell line on the problematic list. For this and all subsequent analyses, we used the edit distance criterion, as it is the measure between the strict and loose criteria and therefore a reasonable compromise. The number of cell lines that could be found on the problematic list was 8.7% (26,418 out of 305,161) of total cell lines, corresponding to 16.1% of papers (*Figure 2*).

The smallest number of cell lines on the problematic list is found in virology and botany related journals, while the largest is in cancer journals. This general observation is consistent

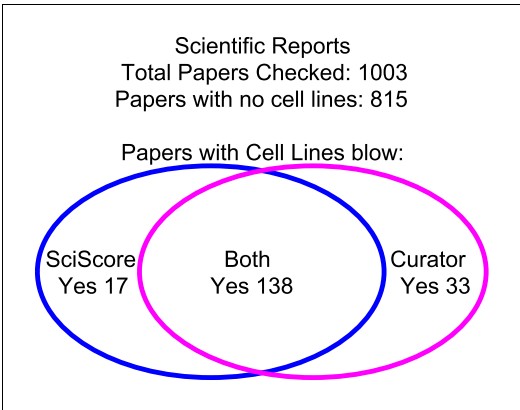

Scientific Reports
Total Papers Checked: 1003
Papers with no cell lines: 815

Papers with Cell Lines blow:

SciScore
Yes 17

Both
Yes 138

Curator
Yes 33

**Figure 3.** The integrity of SciScore for finding papers with cell lines. A manual review of 1,003 papers from the journal Scientific Reports showed a 95% agreement between the curator and the SciScore algorithm. Both the curator and SciScore detected a cell line in 138 articles, and no cell line in 822. Of 1,003 papers, 50 represent a disagreement (false positives and false negatives).
DOI: https://doi.org/10.7554/eLife.41676.006
The following source data is available for figure 3:

**Source data 1.** Underlying data for *Figure 3*.
DOI: https://doi.org/10.7554/eLife.41676.007

with the identity of the problematic list, which has a high proportion of human cancer cell lines. Because of this composition and the aggressive growth of cancer cell lines taking over other cell lines, we expect that cell lines used in cancer journals would be over-represented, while cell lines from model organism would be underrepresented.

To determine the quality of SciScore annotations beyond the training set, we used a single journal, Scientific Reports, that is fully open access (to be read and to be accessed by algorithms). Furthermore, it has a relatively large number of cell line papers as a percentage of the total papers published. To address the issue of false positives and false negatives, we created a manual set of 1,000 papers by first retrieving the subset of Scientific Reports papers via PubMed that span the time frame covered by SciScore, and then selecting every 65th paper, for a total of 1,004 papers. One paper was rejected from the analysis because it was an erratum, leaving 1,003 papers. This method, while not completely random, does sample papers without assuming that the paper will have cell lines. Every methods section was read, and the curator noted the presence or absence of cell lines.

Of the 1,003 papers, both the curator and SciScore did not find any cell lines in 815 papers, but identified cell lines in 138. In total, there was agreement between the algorithm and curator in 953 out of 1003 papers, or 95.01% agreement. Therefore, we did not seek to tune parameters further for SciScore. We assumed that if both curator and SciScore agree about the presence of cell lines in the paper, then the answer is correct and did not look more deeply into these data. We examined the 50 papers with a disagreement to determine whether the algorithm or the curator were incorrect in each case. The curator found 33 papers, which were not found by SciScore, while SciScore found 17 papers that the curator did not.

To look at the false positive rate for SciScore, we examined more closely the 17 papers identified by SciScore but not the curator (*Figure 3*). Of these, the curator was incorrect in 11 out of the 17 cases where SciScore correctly identified a cell line in the paper. For the six cases that were real false positives, the breakdown is as follows: one X-ray crystallography parameter, two reagents, two bacterial strains, and one plant cell line. Thus, the false positive rate was 4% (6/138). The false negatives, or 33 papers that were found by the curator, contained 14 curator errors and 19 legitimate cell lines (*Supplementary file 3*). The overall accuracy of the curator in a cell-line finding task in 1003 papers was: 11 and 14 mistakes out of 1,003, while the algorithm accuracy on the same task was: 6 and 19 mistakes out of 1,003 papers.

### Cell lines identified by RRID
Researchers that publish using the RRID syntax must look up the RRID in a central database (*Figure 4*). This database carries a set of warning messages including the ICLAC warning, echoing the language from Cellosaurus.

We assessed the number of cell lines that were identified by researchers using the RRID syntax, by examining all papers from 2016 to 2017 that contain the term "RRID". Each RRID containing paper was annotated by SciBot, as described in the methods (*Figure 5*). SciBot and/or the curator found and verified 1,554 RRIDs that marked cell lines in 686 papers, 635 of which were associated with a PMID (the unique identifier number used in PubMed). In the list of RRID records, we included those in which the syntax used (e.g., an extra underline or comma) was incorrect, but where the identifier was correct. However, we did not include records where RRIDs were supplied by the

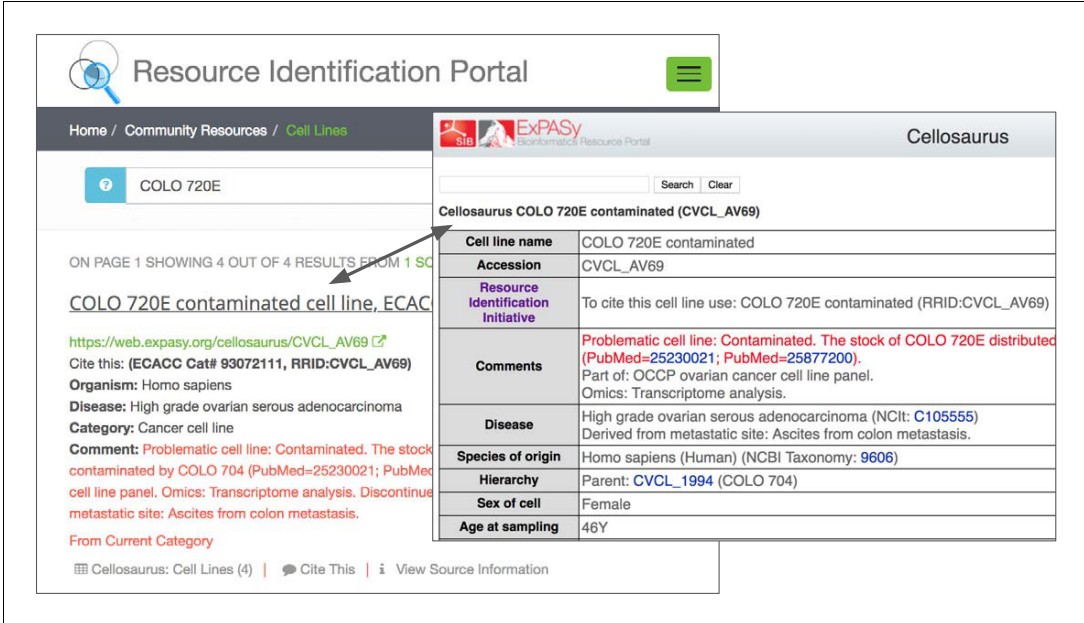

**Figure 4.** The warning message on the RRID portal and the Cellosaurus database present a misidentified cell line, COLO 720E. However, this warning does not originate at either the Cellosaurus (the naming resource for problematic cell lines) or the RRID sites; it simply reflects the information available at ICLAC.org. ICLAC members examine publications and test data to reach a conclusion, and then disseminate this on their website via a spreadsheet available to everyone for download. The Cellosaurus database picks up these data, working closely with ICLAC, and updates their entries. The data are then passed to the RRID portal, where it is displayed for researchers searching for cell lines, among other resources. Cellosaurus and the RRID portal strive to make all new data available as quickly as possible.

DOI: https://doi.org/10.7554/eLife.41676.008

curator and not the researcher. We further trimmed the 1,554 records if the paper was not associated with a PubMed identifier, leaving 1,502 cell lines where researchers used the RRID syntax, published between the year 2016 and 2018.

The number of cell lines identified by RRIDs that matched the list of problematic cell lines obtained from Cellosaurus was 50 out of the 1,502 total cell lines, or 3.3% (*Figure 6*). This 3.3% is significantly lower than 8.7% obtained by the edit distance criterion for all papers (population proportion z test, p<0.00001, z=7.3353). Furthermore, it is also significantly lower than the strict match criterion shown in *Figure 1* (total cell lines 305,130; total strict matches 15,615; or 5.12%; p<0.001, z=3.1406). Limiting the range of dates for the strict criterion to 2016, 2017, and 2018 did not change the average percent of cell lines on the problematic list (total cell lines detected between 2016-18: 110,997; strict problematic: 5,830; 5.25%; p<0.001, z=3.327).

The edit distance, total strict, and the 2016-2018 strict numbers, all are greater than 3.3% at

the p<0.01 significance level. This percentage is also lower than that calculated for any of the individual journals in the top 100 journals (*Figure 6—source data 1*) except 4, including Frontiers in Plant Science (5 cell lines on the problematic list of 399 detected cell lines), Breast Cancer Research: BCR (8 of 423), Nature Structural and Molecular Biology (14 of 253), Cell Reports (12 of 386), and Nature Chemical Biology (17 of 523). These journals may have higher editorial oversight or may predominantly use cell lines that are not listed on the problematic list. This may be especially true for Frontiers in Plant Sciences and Nature Chemical Biology because Cellosaurus and the ICLAC list is skewed toward human cancer cell lines, since the original problem addressed by ICLAC centers on human cancer cell lines.

### Considerations in comparing RRID and open access subsets of literature

PubMed Central currently contains roughly four million papers, and about two million of those are available under a 'text mining is allowed' license. Papers that contain RRIDs are under

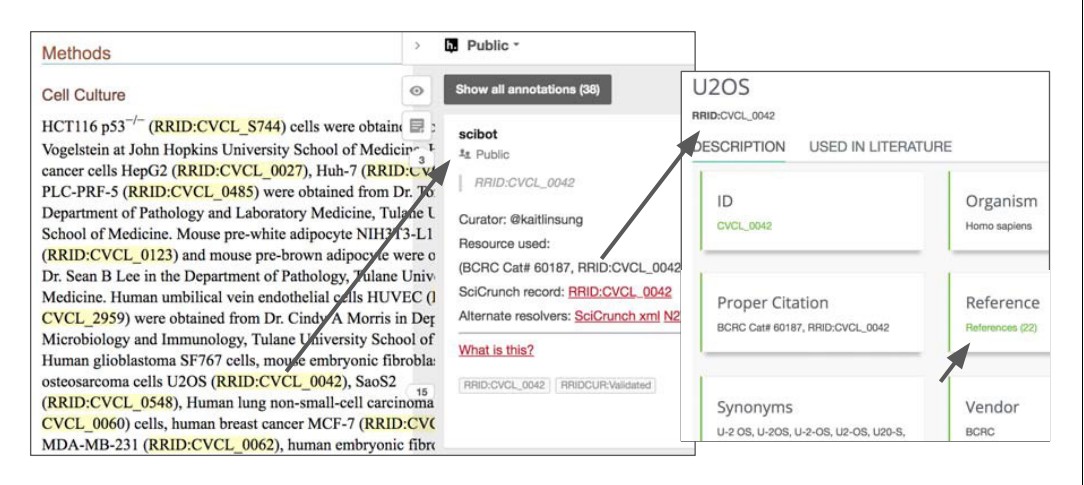

**Figure 5.** An example of a public annotation using the hypothes.is platform. Note, all data made in the public channel, such as RRID resolution data, are ported daily to the CrossRef Event database for developers, providing additional ways of making these data FAIR (that is, findable, accessible, interoperable and reusable). Information about this cell line is accessible to readers with one click, including papers that use the cell line and original reference. For journals like eLife, which typeset the RRIDs with live links, hypothesis is not necessary to access the information about cell lines. Based on the paper *Liao et al., 2017* using the hypothes.is platform. DOI: https://doi.org/10.7554/eLife.41676.009

many different licenses, and only some are available for text mining, so we need to be careful in our comparisons of the RRID literature and the text-minable literature, because these are partially overlapping, but different subsets of the total literature.

We considered if there was a difference in the percentage of cell lines on the problematic list, as we were sampling two different populations of journals. Some journals have higher, and some journals have lower numbers of papers using cell lines on the problematic list (*Figure 2*). We tested the possibility that journals that participate in the RRID initiative have lower rates of use of cells on the problematic list by calculating the weighted average of the journals that have RRID papers and their rate of cell lines present on the problematic list (*Figure 6—source data 1*).

The rate of contamination was gathered from the data underlying *Figure 3* (*Figure 3—source data 1*). Briefly, the edit distance metric is used to match whether the cell line matches a cell line on the problematic list and the data are provided as percentages by journal. Eighty-eight journals contained cell line RRIDs and eight of these journals and a total of 37 papers did not have matching data in the open access set. The latter include 30 papers from The Journal of Neuroscience and seven papers from various journals that are not open to text mining.

We used a weighted average to determine what the rate of possible cell line contamination for each journal and found that the average was 17.1%, a value slightly larger than the overall average of 16.1%. Therefore, we do not believe that the journals using RRIDs are simply better than others in terms of cell line authentication. In fact, inclusion of an RRID is a more likely due to an intervention than a journal policy that reduces the overall use of cell lines on the problematic list.

However, the presence of a cell line on the misidentified list in a paper does not mean that the statements or conclusions of the paper are questionable. In fact, many of the cell lines that are on the problematic list are explained by a mix up between cells originating from patients suffering from the same disease as the patient who was supposed to be the donor of the contaminated cell line, which is not something that should impact the conclusions of a study. Indeed, researchers may knowingly use a hepatocarcinoma cell line even if it may have been mislabeled as a hepatoblastoma, because their experiment calls for a hepatocarcinoma cell.

Moreover, sometimes, it may not be necessary to know the exact origin of the cell line as their experiment simply calls for mammalian cells, especially when a cell line is only used as a vector for (over)expressing a specific gene. While it is not practical to look at thousands of

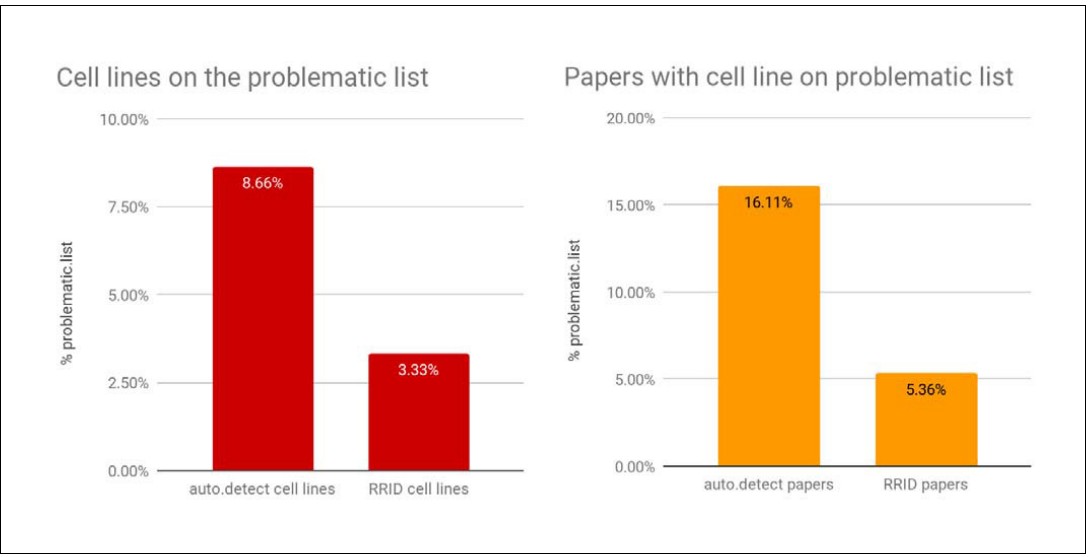

**Figure 6.** Percentages of papers with cell lines found on the problematic list. The "auto.detect.cell" lines data come from the edit distance metric, same as *Figure 2*; n=305,161; the RRID cell lines are based on 1,502 cell lines. The "auto.detect" papers percentage is based on n=150,459 unique papers, where the problematic cell-line list is detected based on the edit distance metric. The RRID papers percentage is based on n=634 papers.
DOI: https://doi.org/10.7554/eLife.41676.010
The following source data is available for figure 6:

**Source data 1.** Data on number of misidentified cell lines per year.
DOI: https://doi.org/10.7554/eLife.41676.011

papers to determine if the researchers knew that they were working with a misidentified cell line, it was practical to look at the 50 cell lines used in the RRID literature because presumably, they should have been alerted of a potential problem with their cell line.

Here, we found that one out of the of 50 cell lines on the list where an RRID was included by the researchers was a typo – they clearly used an uncontaminated cell line, but transposed the Cellosaurus number incorrectly. But 13 carry the 'partially contaminated' label, because the SH-SY5Y cell line (RRID:CVCL_0019) was contaminated in a laboratory separate to the originating laboratory, and several other laboratories who obtained the cell line used the contaminated version.

Of these 13 papers, 11 clearly state that the cell line was obtained directly from a stock center such as ATCC, which contains an authentic version of the cell line. Ten papers used the U-87MG ATCC cell line (RRID:CVCL_0022) originally published as a glioblastoma cell line established in 1968 at the University of Uppsala, but this cell line is most likely a glioblastoma cell line of unknown origin. Therefore, as far as we can tell there is no cause for concern in these studies.

The cell line Hep-G2 was used in 11 papers. Hep-G2 was originally thought to be a hepatocellular carcinoma cell line, but they are now confirmed to be hepatoblastoma cells. In these papers, four papers assert they used a hepatocellular carcinoma cell, while the other seven used more generic terminology, such as "liver cancer cell line", suggesting that the cell line origin was relatively irrelevant to the conclusions.

The remaining cell lines break down into many classes, and in roughly half of the cases, the researchers seem to be using terminology consistent with the true identity of the cell. In the other half, they refer to the cell line as the original, now known to be an untrue cell line (see *Supplementary file 4* for a detailed account). Overall, only about 10 of the cell lines warrant a more in-depth look to see whether they affected the conclusions of the paper.

## Discussion

Despite high-profile articles detailing the problem with cell contamination and misidentification, papers using cell lines on the ICLAC register of misidentified cell lines continue to be published (*Horbach and Halffman, 2017*). Scientists do not yet routinely use short-tandem-

repeat profiling to test for contamination. Such practices are one reason why in 2015, the National Institutes of Health (NIH) adopted stricter guidelines for authenticating key biological reagents, including cell lines (*NIH, 2015*). In this study, we presented data suggesting that the efforts of ICLAC to make information about misidentified cell lines more easily accessible, appear to be having an effect on the trajectory of the use of misidentified cell lines. Moreover, adding RRID syntax into an unpublished manuscript and forcing researchers to look at the database record for that cell line, is correlated with a decrease in the reporting of problematic cell lines.

It is also linked to less reports of the use of problematic cell lines in our sample of over 1,500 cell lines from over 600 papers. The RRID portal, to which researchers are directed by their journal, carries a warning message for all problematic cell lines, which we recently turned red to make it more obvious (*Figure 4*). To obtain the RRID, scientists can look at this entry either on the RRID portal or on the Cellosaurus portal, both of which show the same warning about the cell line. It stands to reason that researchers, who see this warning are likely to consider the implications for their study and act in good faith. However, it is also possible that they may simply copy the RRID from another paper, and therefore, would not see the warning. While we are unable to determine what percentage finds their RRIDs this way, we consider it an important problem.

Our assessment of the problematic cell-line use is based on whether the culture shows up on the ICLAC register of misidentified cell lines or as problematic on the Cellosaurus database. Although the ICLAC list can alert researchers to the true identity of a cell line, it cannot provide a value judgment regarding the use of that cell line in their study. Scientists may use a cell line that was once thought to be a hepatoblastoma, but is now known to be a hepatocarcinoma in an experiment to test hepatocarcinoma cells, i.e., the proper use of this cell line, or they may discuss the cell line as a hepatoblastoma, the wrong use. They may also be testing a general property of cancer cell lines and the conclusions are not affected by the specific subtype of cancer, and just refer to them as "liver cells". Therefore, the current estimates for the use of problematic cell lines are not a definitive statement that 8.6% of cell lines from 16.1% of papers are in need of being reviewed, as many of these may be using the cell lines properly.

However, this estimate is better than previous attempts based on literature search strategies.

The frequency of misidentified cell lines in the research community depends on its source and whether it has been authenticated; cell lines from primary sources (originator or reference repository) are less likely to be misidentified compared to secondary sources (*Drexler et al., 2017*). Estimates of misidentified cell lines in the research community vary from around 10% (*Liang-Chu et al., 2015*) to close to 50% (*Schweppe et al., 2008*; *Huang et al., 2017*). It has been suggested that tens of thousands of cell line papers should be revisited as a result of the likely use of misidentified cell lines (*Neimark, 2015*; *Zhao et al., 2011*; *Masters, 2012*). Estimates of affected publications are largely based on search strategies using PubMed or PubMed Central (*Vaughan et al., 2017*; *Horbach and Halffman, 2017*). The number of total papers is so large that looking at each case manually would be a daunting task.

However, while search strategies are a great way to help identify papers that use misidentified cell lines, they have several limitations. Many names, especially those for the oldest and thus most established experimental cell lines, are short (e.g., "KB"), use generic words (e.g., "Chang liver"), or are provided in abbreviated form. It can be difficult to know if the name is referring to a gene or a part of a figure as opposed to a cell line. Here, we describe our use of text mining to look for sentences in the methods sections of open access papers that describe cell lines, based on sentence patterns and text features around the cell line name.

Our approach for finding cell lines using automated detection in the PubMed Central corpus has significant limitations. Importantly, an uneven number of papers per year, including a partial year (2018), is likely to affect the accuracy of counts, especially in the older literature, where there are simply fewer papers per year. We report data as a percentage of total, yet this decision in a sense, hides the fact that the numbers of papers per year is different. Indeed, we see a larger variability year to year in early years as opposed to later years (*Figure 1*).

Older papers – especially those before the year 2000 – which were entered into PubMed Central in a more one-off or manual manner from years prior to the creation of this database, may be subject to more errors than current papers that are processed at the publishers and sent as structured data. We have no good estimate about the proportion of errors in newer or

older literature. While issues with character recognition would presumably creep into the subset of manually handled papers more frequently, SciScore should still recognize the word as a cell line, regardless of whether the word matches a particular cell line name. However, if a character error creeps into the cell line name, it would be less likely to match a problematic cell line, given our fairly stringent criteria for problematic cell line identification.

Our understanding of cell lines continues to change, and this must be taken into consideration alongside the RRID. Inclusion in the ICLAC register of misidentified cell lines is based on published data, available samples, and existing test methods. New data may lead to cell lines being added or removed from the register, based on a discovery of authentic material or a new finding of misidentification. Similarly, information on problematic cell lines in Cellosaurus is based on available evidence and may change over time. Because RRIDs make it possible to search easily for papers that mention a particular cell line, it may be possible to use literature annotation tools like Hypothes.is to provide updated information on a published paper when new information becomes available.

It is important to review the evidence on which these findings are based and make a considered judgment regarding their impact on published work. Cell line resources, such as Cellosaurus and the ICLAC register of misidentified cell lines, have been developed to improve awareness of cell line information. The inclusion of RRIDs will result in improved use of these resources and better reporting of cell lines and other research materials in publications.

## Methods

### Text mining for cell line mentions in the open access subset of PubMed Central

To extract mentions of cell lines in papers, we utilized SciScore, under development by SciCrunch Inc (RRID:SCR_016251). SciScore is a text analysis tool suite that uses Named Entity Recognition to extract words or named entities from text documents. Here, we focused on words indicating cell lines. A word like HeLa is relatively unambiguous, however this is not the case for many other cell-line names. Therefore, the SciScore algorithm works only partially by recognizing these names, and mainly identifies them based on the sentence context of the word. SciScore is thus capable of finding new

cell-line names that it was never specifically trained to recognize. This approach has been described previously (*Ozyurt et al., 2016*).

In the current experiments, SciScore was instructed to extract the methods section of each paper, and then find names of cell lines within (*Box 1*). This way, SciScore searches for papers that are more likely to have used a cell line as opposed to papers that simply discuss a cell line. To create a training dataset for SciScore, 1,457 cell lines were annotated by two human curators, with a high inter-curator agreement (over 90%). The human-curated data were split into datasets for training (90%) and testing (10%). SciScore was trained on ten different 90/10 splits. The average values for precision, recall, and the harmonic mean of precision and recall (F1-score) are reported. The detected values were calculated according to standard definitions from the text mining literature, also described in the statistics section, below.

SciScore ran on the open access subset of PubMed Central (RRID:SCR_004166), retrieved on [3/23/2018], which constituted 1,950,740 total papers published between January 1975 and March 2018. SciScore identified 673,272 cell-line names in the methods section of 150,459 unique papers. According to this analysis, 7.7% of papers in the open-access subset of PubMed Central use cell lines. In those, we found 305,161 unique cell-line names in total. Therefore, in our sample of about 150,000 papers, each methods section describes an average of two cell lines and each cell line is mentioned twice.

To determine SciScore performance in finding cell-line papers, we tested SciScore against an independent human-curated set of data. We anticipated that we would need about 100 examples of papers that contained cell lines, but the sample of all papers shows that cell-line papers represent a relatively low proportion of all papers (~7.7%). To improve our chances of finding 100 or more exemplary papers, we selected Scientific Reports, a journal that publishes many cell line papers and also is fully open access. The curator searched PubMed and downloaded the basic metadata (author names, dates and title for each paper in Scientific Reports published between the available dates 06/14/2011 and 04/01/2018) were chosen. The curator was blinded to the output of the algorithm while curating papers.

Briefly, after searching PubMed for all papers published in Scientific Reports, we downloaded a CSV file from PubMed containing the basic

## Box 1. Example sentences containing cell lines detected by the SciScore algorithm.

The following two sentences were annotated both by curators and by SciScore. The highlighted text corresponds to the words that represent cell lines found by curators or by SciScore. Example results of SciScore are shown in parantheses (correct annotations, false positive: incorrectly identified a plasmid as a cell line, false negative: failed to find a cell line that the curator found).

**Sentence 1** (methods sentence line 353; PMID:26012578)

For luciferase activity assays, **HeLa** (correct annotation) or **HCN-A94** (correct annotation) cells were grown in 24 well plates and transfected with 0.1 µg **phRL-TK-10BOXB** (false positive) plasmid, 0.1 µg of pGL3 promoter plasmid and with 0.7 µg of one of the six pCl- λN-HA-tagged UPF3B expression constructs.

**Sentence 2** (methods sentence line 125; PMID:28638484)

For cellular uptake kinetics study, **HeLa** (false negative) or **RAW264.7** (correct annotation) cells were seeded into 96-well plates and allowed to attach for 24h.

DOI: https://doi.org/10.7554/eLife.41676.012

metadata for all these papers. Review and non-research papers lacking an abstract were removed, resulting in a total of 65,085 articles for analysis. Of these, every 65th paper, by date of publication, was selected for curatorial review, leaving 1,004 papers to be reviewed for cell line presence. Each of the these was then manually reviewed by a curator to determine if cell lines had been used and if so, which ones. One record was an erratum and was removed from this final list, leaving 1,003 papers to examine.

### The problematic cell-line list

We extracted a list of problematic cell lines from Cellosaurus database Version 25 (March 2018), and copied the 810 cell line identifiers, 1,811 names and synonyms into a document (**Supplementary file 2**). The composition of this list is described in the Cellosaurus Frequently Asked Questions section and contains cell lines from the ICLAC Register of Misidentified Cell Lines (both approved cell lines and those submitted for ICLAC review) and additional problematic cell lines reported by other sources. For the purposes of our primary analysis, we did not distinguish between different categories of cell line problems. However, we considered the differences in a subsample of the RRID papers, where it was important to determine whether the researchers knowingly used a misidentified cell line and if so, if they knew the identity of it.

To determine if cell-line names were highlighted in Cellosaurus as misidentified or otherwise problematic, we matched the data we obtained from SciScore, with the extracted list using three methods: strict, loose, and edit distance. The strict method involves loading the data into excel and using the "vertical lookup" function in Excel (VLOOKUP) to match whether the exact name for a cell line found by SciScore was one of the names or synonyms. For the loose method, we added an asterisk * to the end of each cell line name to match additional possibilities. The strict and loose methods for matching names represent what we consider the outer boundaries as percentages for a cell-line presence on the problematic list. Excel treats matches in a case-insensitive manner, so "HeLa" and "HELA" would both be considered a match.

The third method used was the edit distance method to solve the string-to-string correction problem (**Wagner and Fischer, 1974**). The minimum edit distance is the minimum number of character deletion, addition and substitution operations necessary to make two strings equal. We have used weighted edit distance where the deletion or addition of certain special characters had 0 weight, other deletion and additions had a weight of 1. Substitutions had a weight of 2 (a deletion and addition). The match was considered valid if the edit distance was less than 1.

### Corpus of papers containing RRIDs

We searched for all papers that contain RRIDs by monitoring Google Scholar, ScienceDirect (Elsevier), Wiley and PubMed Central every week for four years for mentions of the string RRID. Each paper that contained the term "RRID" was opened by a curator and listed in our database. For the past two years, each paper was examined by our semi-automated tool, SciBot (*Gillespie and Udell, 2018*; RRID: SCR_016250; copy archived at https://github.com/elifesciences-publications/scibot) for RRID mentions. SciBot is an annotation-based curation workflow that works with the Hypothesis annotation tool (Hypothes.is; RRID:SCR_000430). The curation workflow comprises the following steps:

1. Document identification: for each paper, SciBot first identifies the digital object identifier (DOI) and the PubMed identifier (PMID) by locating the DOI on the html webpage and then querying the PubMed application programming interface to respond with a PMID for the specified DOI. These identifiers are attached to the document URL through the Hypothesis annotation service. If the automated methods fail, the curator fills in the PMID using the Hypothesis annotation client.

2. RRID recognition and resolution: SciBot then scans the text of the paper and finds all places in the document that appear to be RRIDs by searching for the string "RRID". The text directly after the RRID and before the next break, such as a comma, space or period, is then submitted to the Scicrunch resolving service, the results of which are then provided to the curator as a linked annotation viewable in Hypothesis (*Figure 5*). If there is no RRID record, SciBot returns an annotation "Unresolved" tag. If the RRID is resolved, the curator verifies whether the researcher and the database record agree or disagree. Curators also note whether there are any errors, such as incorrect records, and note whether resources are referenced that should have received an RRID that did not ("Missing").

3. Data export: The data from SciBot are stored as a JSON file in Hypothesis and are then extracted to the SciCrunch annotation database, where the paper is assigned to each RRID that it references, along with the assigned curation tags (e. g., "Verified").

RRID Dataset: The RRID dataset as of March 2018 included over 2,000 cell line annotations. RRIDs that were supplied by the curator ("Missing") were excluded from this data set, leaving 1,554 cell lines where the researcher had added the RRID. Of the 1,554 cell-line RRIDs, 1,502 cell lines were found in papers associated with a PMID. The data set as of 4/01/2018 contains 686 unique papers; 634 of these were associated with a PMID. Papers without a PMID were not considered in the paper-level analysis, as we could not guarantee that they were unique.

### Statistics

To determine if the RRID literature and the general literature represent distinct populations, we used the population proportion z-score statistic to deal with the difference between the total number of cell lines reported in papers and the cell lines reported via RRID (Social Science Statistics online calculator, RRID:SCR_016762). We made the one-tailed assumption as we assumed that researchers would only decrease their use of problematic cell lines and no longer publish data generated using problematic cell lines.

For SciScore, we used standard measures for classification performance: precision P, recall R, and harmonic mean of precision and recall F1. These are defined by the following formulas: the number (#) of correctly recognized labels refers to the number of words that are recognized as cell lines, which were also a cell line according to the curators. The number (#) of true good labels refers to the total number of cell lines according to the curators. P, R, and F1 are calculated on each 10% test set.

$P = $ (# of correctly recognized good labels) / (# of recognized good labels)

$R = $ (# of correctly recognized good labels) / (# of true good labels)

$F1 = (2*P*R) / (P+R)$

### Acknowledgments

We thank the following individuals for their contribution to this work: Joseph Menke, who worked tirelessly in curating cell lines and RRIDs in the published literature; Jon Udel, who wrote the original SciBot and continues to have the vision to bring the future into the present; Edyta Vieth, who works with the RRID literature and contributed helpful insights in conceiving this manuscript. We also thank Dr Anthony C. Gamst for checking our statistical assumptions. Moreover, we thank the many journal editors and editorial staff who continually work to improve the research integrity by augmenting their workflow to include RRIDs.

**Zeljana Babic** is in the Center for Research in Biological Systems, University of California, San Diego, United States

**Amanda Capes-Davis** is in the Children's Medical Research Institute, University of Sydney, Westmead, Australia

**Maryann E Martone** is in the Department of Neuroscience, University of California, San Diego, United States and SciCrunch Inc., San Diego, United States

**Amos Bairoch** is in the Computer and Laboratory Investigation of Proteins of Human Origin, Swiss Institute of Bioinformatics and the Department of Microbiology and Molecular Medicine, University of Geneva, Geneva, Switzerland

**I Burak Ozyurt** is in the Center for Research in Biological Systems, University of California, San Diego, United States

**Thomas H Gillespie** is in the Neurosciences Graduate Program, University of California, San Diego, United States
https://orcid.org/0000-0002-7509-4801

**Anita E Bandrowski** is in the Center for Research in Biological Systems, University of California, San Diego, United States and SciCrunch Inc., San Diego, United States
abandrowski@ncmir.ucsd.edu
https://orcid.org/0000-0002-5497-0243

*Author contributions:* Zeljana Babic, Data curation, Validation, Methodology, Writing—original draft, Writing—review and editing; Amanda Capes-Davis, Resources, Validation, Writing—review and editing; Maryann E Martone, Resources, Funding acquisition, Writing—original draft, Project administration, Writing—review and editing; Amos Bairoch, Resources, Data curation, Validation, Writing—review and editing; I Burak Ozyurt, Software, Visualization, Methodology, Writing—review and editing; Thomas H Gillespie, Conceptualization, Software, Methodology, Writing—review and editing; Anita E Bandrowski, Conceptualization, Supervision, Funding acquisition, Validation, Writing—original draft, Project administration, Writing—review and editing

*Competing interests:* Amanda Capes-Davis: is a Cell-Bank Australia Honorary Scientist and Chair of ICLAC, both on a voluntary basis. Maryann E Martone, Anita E Bandrowski: heads the RRID project, and founded Sci-Crunch, a company that supports the RRID project. Amos Bairoch: develops the Cellosaurus database. I Burak Ozyurt: works as a consultant for SciCrunch. The other authors declare that no competing interests exist.

**Funding**

| Funder | Grant reference number | Author |
|---|---|---|
| NIH Office of the Director | OD024432 | Anita E Bandrowski |
| NIH Blueprint for Neuroscience Research | DA039832 | Maryann E Martone |
| National Institute of Diabetes and Digestive and Kidney Diseases | DK097771 | Maryann E Martone |
| National Institutes of Health | MH119094 | Maryann E Martone Anita E Bandrowski |

The funders had no role in study design, data collection and interpretation, or the decision to submit the work for publication.

**Decision letter and Author response**
Decision letter https://doi.org/10.7554/eLife.41676.020
Author response https://doi.org/10.7554/eLife.41676.021

## Additional files
### Supplementary files
• Supplementary file 1. List of RRID papers.
DOI: https://doi.org/10.7554/eLife.41676.013

• Supplementary file 2. List of problematic cell lines extracted from Cellosaurus Version 25 (March 2018).
DOI: https://doi.org/10.7554/eLife.41676.014

• Supplementary file 3. Curator-SciScore-disagreement - the false negatives (33 papers) found by the curator.
DOI: https://doi.org/10.7554/eLife.41676.015

• Supplementary file 4. Details for the 50 RRID papers.
DOI: https://doi.org/10.7554/eLife.41676.016

• Transparent reporting form
DOI: https://doi.org/10.7554/eLife.41676.017

### Data availability
The data underlying the figures and discussions is available as a set of supplementary files to this manuscript. However, as this project is ongoing the data from this manuscript and subsequently curated data is available from scicrunch.org for individual records (human readable example https://scicrunch.org/resolver/RRID:CVCL_0058; machine readable example https://scicrunch.org/resolver/RRID:CVCL_0058.xml ) and from Crossref's Event database in bulk (example query https://api.eventdata.crossref.org/v1/events?rows=10000&source=hypothesis&obj-id.prefix=10.1016 ).

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
