## [Decision Letter]

Thank you for submitting your article "Incidences of problematic cell lines are lower in papers that use RRIDs to identify cell lines" to *eLife* for consideration as a Feature Article. Your article has been reviewed by three peer reviewers, and the evaluation has been overseen by two staff editors. The following individuals involved in review of your submission have agreed to reveal their identity: Malcolm R MacLeod (Reviewer #1) and Christopher Korch (Reviewer #3). Reviewer #2 remains anonymous.

We invite you to submit a revised version of your manuscript that addresses the points made by the reviewers (which are summarised below).

Summary:

Using misidentified cell lines for biomedical research is a problem that has persisted long after it was identified in the 1960s and highlighted by the work of Stanley Gartler, Walter Nelson-Rees, many others. The frequency of misidentified cell lines in various collections of cell lines ranges between 10% -100%, depending on the sample size and the origin of the tissue. This persisting problem probably causes many irreproducible results.

This article describes a large-scale survey of the scientific literature and aims to see if using Resource Reference Identification tags (RRIDs) to describe cell lines in research articles has reduced the use of misidentified cell lines, for which no known authentic samples exist. The authors found that using RRIDs reduced the incidence of misidentified cell lines to about 40%, compared to articles that did not use RRIDs. This major decrease illustrates a hopeful approach to tackling the use of known misidentified cell lines.

Essential revisions:

a) Materials and Methods section: Was the study design articulated in a study protocol, and is a dated version of this available? That is, can the authors demonstrate that their primary measures of interest and statistical analysis plan were articulated prior to data collection? If not, can you give some indication of any other analyses that were performed but are not reported? It might be worth listing these (if there are any).

b) Results section: Please report the results of all analyses in the Results section – it is not sufficient to state that "all are different".

c) The ICLAC register of misidentified cell lines consists of two lists. Table 1 is a list of misidentified cell lines for which there does not exist any known authentic samples. Table 2 is a list of misidentified cell lines, but for which there exist authentic samples. Please clarify which subset of the ICLAC register was used in this study?

d) The Discussion section does not cover some important aspects of the study, such as:

- The inclusion of a partial calendar year (2018) in the analysis and how this may have affected the results.

- The predictably low use of cell lines (in a searchable format) in older papers and how this may have affected the resultant correlation coefficient.

- The 14% miss rate (false negatives – subsection “Text mining corpus”) – this seems a rather high number given the human assessment is presented as the standard against which the automated search is compared.

- There is a rather large assumption that researchers check the cell database early in a project and would cease to use problematic cells if a warning is seen (subsection “Cell Lines identified by RRIDW”, Discussion section). Evidence to support this assumption is missing.

- Was the potential for older scanned papers to not be of sufficient resolution/ quality for text mining considered? If so, how could this have affected the results?

e) Discussion section. In this paragraph the authors mention the range of 15-30%. It seems that they do not understand that in the two references to which they refer for these values, these numbers are the range of frequencies of misidentified or cross-contaminated cell lines in collections, but are not frequencies of the number of publications using misidentified cell lines. Moreover, the Masters, 2012 reference is a letter to Nature that refers to an editorial in Nature from 2009, and not to a study that actually determined these values. This frequency information is used incorrectly in the context of this paragraph that discusses search strategies for identifying published articles that have used misidentified cell lines.

Furthermore, the frequency of misidentified cell lines ranges more broadly, between 10% and 100%, depending on the sample size and type of cell lines. For the frequency of misidentified cell lines in collections of cell lines I would suggest referring to instead Drexler et al., 2003 (18-36%) and Zhao et al., 2011 (30%). Perhaps one could also include more recent studies that show higher frequencies, some close to 50% (Schweppe et al., 2008, 48% of thyroid lines; Huang et al., 2017, 46% of various lines). At the lower end, there is Liang-Chu et al., 2015, who observed in a large collection a frequency of 9.6% in which they compared many different cell lines. Based on these ranges, a more meaningful range of the frequency of cross-contaminated or misidentified cell lines in collections would be 10% to 48%.

This information could be used as the first sentence in the paragraph to say how frequently misidentified cell lines are found. Then followed by referring to the Neimark, 2015 reference, and followed by a reference to other recent studies. For references to published articles that have examined the frequency that misidentified cell lines have been used in published studies (which would be more appropriate for this paragraph), I suggest weaving in the story of KB cells by Vaughn et al., 2017 and the findings of Horbach and Halffman, 2017. These authors used different strategies for identifying publications that used misidentified cell lines, which is the topic of this paragraph.

f) Supplementary data: It's not clear whether the Supplementary data are a summary (it appears to be) or the entire dataset. I know that the entire dataset is evolving and available, but if what is presented here is a summary then I'd prefer that you deposited a snapshot of the data at the time the analysis was done, to allow attempted replication using the exact same dataset. But I may have got this wrong.

g) Although beyond the scope of this study, the use of RRIDs does not verify that a cell line used for a study is indeed the claimed cell line. That would require each lab to authenticate the samples of the cell lines they are using by STR or SNP genotyping. If journals and funding agencies would enforce this, it may help to prevent misidentified cell lines from being unknowingly used in biomedical research. Please consider adding a sentence or two on this topic.

---

## [Author Response]

Essential revisions:a) Materials and Methods section: Was the study design articulated in a study protocol, and is a dated version of this available? That is, can the authors demonstrate that their primary measures of interest and statistical analysis plan were articulated prior to data collection? If not, can you give some indication of any other analyses that were performed but are not reported? It might be worth listing these (if there are any).

We certainly should have registered this study and failed to do so. This was a mistake. However, the idea originated in this Blog, published on May 11, 2017: https://elifesciences.org/inside-*eLife*/ff683ecc/rrids-how-did-we-get-here-and-where-are-we-going

At the time of the blog, there were only about 100 papers using RRIDs to tag the cell lines used in the paper and we had two examples of problematic lines in that set, a percentage that seemed quite small. We did not write a full paper because the data set was preliminary and no comprehensive set of data existed for use of misidentified or contaminated cell lines in the broader literature. We subsequently created a larger data set in the course of creating text mining algorithms to identify research resources in the literature (this work was not undertaken to conduct this study, but to create a tool for editors, which we are currently β testing). When the algorithm training was performing sufficiently for our needs, we ran it once over the total biomedical literature, which we routinely parse for other reasons, to gather the data for this paper. Within a day of the algorithm run, we took all RRID data and put it into a file to analyze and did not re-curate or fix any of it. The papers described in this manuscript as lacking a PubMedIDs are almost guaranteed to now have those in our systems, and the RRID data today (11/8/2018) includes 4,018 cell lines because this is a rapidly growing data set. The use of the population proportions z test statistic was done after consulting with a UCSD statistician, Dr. Anthony C. Gamst, to make sure that we were not making a mistake regarding the type of statistical test that we used. We did no analysis on these data other than what is reported in the paper. We did retest after review, using an online social statistics calculator (now cited) to verify the z scores in a place easily accessible to readers.

b) Results section: Please report the results of all analyses in the Results section – it is not sufficient to state that "all are different".

We have added the following language in subsection “Considerations in comparing RRID and open access subsets of literature”, which includes an online calculator where the numbers can be plugged in to verify our claims and the exact z score obtained:

“This 3.3% is significantly lower than 8.7% obtained by the edit distance criterion for all papers (population proportion z test, https://www.socscistatistics.com/tests/ztest/Default2.aspx, p<0.00001, z=7.3353). Furthermore, it is also significantly lower than the strict match criterion shown in Figure 1 (total cell lines 305,130; total strict matches 15,615; or 5.12%; p<0.001, z=3.1406). Limiting the range of dates for the strict criterion to 2016, 2017, and 2018 did not change the average percent of cell lines on the problematic list (total cell lines detected 2016-18: 110,997; strict problematic: 5,830; 5.25%; p<0.001, z=3.327). Whether we consider the edit distance, total strict, or the 2016-2018 strict numbers, all are different from 3.3% at the p<0.01 significance level.”

- We note that the statistical calculator actually states that “The Z-Score is 7.3353. The p-value is 0.” for the edit distance metric, however we did not think that it would be wise to say that our p is zero, we therefore used the p<0.00001 as an approximation.

c) The ICLAC register of misidentified cell lines consists of two lists. Table 1 is a list of misidentified cell lines for which there does not exist any known authentic samples. Table 2 is a list of misidentified cell lines, but for which there exist authentic samples. Please clarify which subset of the ICLAC register was used in this study?

We updated the Discussion section to address this question. In short, the list of cell lines which were flagged as problematic in this paper are the ones that have a "CC" comment line "Problematic cell line" in the Cellosaurus. The text now reflects the link to Cellosaurus database FAQ and the paragraph copied below:

“We extracted a list of problematic cell lines from Cellosaurus version Version 25 (March 2018), and copied the 810 cell line identifiers, 1,811 names, and synonyms, into a document (Figure.Data file, worksheet “Problematic.List”). The composition of this list is described in the Cellosaurus Frequently Asked Questions section (https://web.expasy.org/cgi-bin/cellosaurus/faq#Q17) and contains misidentified cell lines from the ICLAC Register of Misidentified Cell Lines, and cell lines as well as additional problematic cell lines reported by other sources. It must be noted that this list contains both ICLAC approved cell lines and those that have been submitted for ICLAC review.”

d) The Discussion section does not cover some important aspects of the study, such as:

We have added a paragraph in the Discussion section to address some of these points. We also address specific concerns here.

- The inclusion of a partial calendar year (2018) in the analysis and how this may have affected the results.

It is relatively unlikely that the inclusion of the partial calendar year has seriously affected the results. While the absolute number of papers for 2018 is indeed smaller than 2017, it is still larger than most of the early data prior to 2012. There is no valid reason that we can think of to suggest that papers published early in a year would use more or less problematic cell lines.

- The predictably low use of cell lines (in a searchable format) in older papers and how this may have affected the resultant correlation coefficient.

PubMed Central started around the year 2000 so there are few papers in the years prior to 2000 as shown in figure 1 and in the data file. We therefore can be less confident of the total number of cell lines used in earlier literature because our sample is much smaller. However, the open access subset of the literature, as it is added to PMC, is consistently coded in JATS XML format by the National Library of Medicine staff as the papers are processed, which is true for new and early papers. Papers, regardless of age, that are only available as PDFs are not visible to our robots therefore are excluded from our sample. Therefore, it is highly unlikely that recognition of cell lines is substantially different in the older vs more recent literature. Typographical errors can always creep in, but we do not use optical character recognition or any other mechanism to gather papers that are not formatted by professionals

- The 14% miss rate (false negatives – subsection “Text mining corpus”) – this seems a rather high number given the human assessment is presented as the standard against which the automated search is compared.

We believe that this point was unclear in our writing and we have added the percent error into the relevant text. The curator did make 14 errors out of the total 1003 papers or 1.4%, which is certainly far from perfect, but not 14%. Subsection “Cell lines identified by RRID” was updated to include the percentage.

- There is a rather large assumption that researchers check the cell database early in a project and would cease to use problematic cells if a warning is seen (subsection “Cell Lines identified by RRIDW”, Discussion section). Evidence to support this assumption is missing.

We thank the reviewer for pointing this out, we have removed some of the excessive “hand waving” language from the Discussion section. We have no evidence for why authors report fewer cell lines. We also changed the sentence structure in subsection “Considerations in comparing RRID and open access subsets of literature” to ensure that the only thing we know is that the author must find the number at some point during the study and the journals instructions direct them to the RRID portal.

- Was the potential for older scanned papers to not be of sufficient resolution/ quality for text mining considered? If so, how could this have affected the results?

The text mining algorithm uses the data in xml format directly from PubMed Central, and while it was certainly seeded with the names of all cell lines, it is not bound to only recognize those names. The algorithm recognizes the context of the cell line in the sentence, including the most significant sentence features such as other noun phrases surrounding the putative cell line and the structure of the sentence. This allowed the algorithm to find cell lines that are not in the Cellosaurus database, and the more commonly found cell lines detected by the algorithm were subsequently added to Cellosaurus. It is certainly possible that transcription errors from PDF files submitted to the National Library of Medicine are present and many more of these would be expected in the older literature, however the detection of cell lines should not be affected. If errors did creep into the core documents, we would expect that cell lines that were on the problematic list would likely be affected because while a single character change should still trigger the algorithm to detect the cell line, but our analysis of how close the cell lines are to the list of problematic cell lines would be potentially no longer a “zero” edit distance from a problematic cell line.

e) Discussion section. In this paragraph the authors mention the range of 15-30%. It seems that they do not understand that in the two references to which they refer for these values, these numbers are the range of frequencies of misidentified or cross-contaminated cell lines in collections, but are not frequencies of the number of publications using misidentified cell lines. Moreover, the Masters, 2012 reference is a letter to Nature that refers to an editorial in Nature from 2009, and not to a study that actually determined these values. This frequency information is used incorrectly in the context of this paragraph that discusses search strategies for identifying published articles that have used misidentified cell lines.Furthermore, the frequency of misidentified cell lines ranges more broadly, between 10% and 100%, depending on the sample size and type of cell lines. For the frequency of misidentified cell lines in collections of cell lines I would suggest referring to instead Drexler et al., 2003 (18-36%) and Zhao et al., 2011 (30%). Perhaps one could also include more recent studies that show higher frequencies, some close to 50% (Schweppe et al., 2008, 48% of thyroid lines; Huang et al., 2017, 46% of various lines). At the lower end, there is Liang-Chu et al., 2015, who observed in a large collection a frequency of 9.6% in which they compared many different cell lines. Based on these ranges, a more meaningful range of the frequency of cross-contaminated or misidentified cell lines in collections would be 10% to 48%.This information could be used as the first sentence in the paragraph to say how frequently misidentified cell lines are found. Then followed by referring to the Neimark, 2015 reference, and followed by a reference to other recent studies. For references to published articles that have examined the frequency that misidentified cell lines have been used in published studies (which would be more appropriate for this paragraph), I suggest weaving in the story of KB cells by Vaughn et al., 2017 and the findings of Horbach and Halffman, 2017. These authors used different strategies for identifying publications that used misidentified cell lines, which is the topic of this paragraph.

We thank the reviewers for such a thorough evaluation of the background material and the many helpful suggestions about more appropriate references. We have made changes to the manuscript to reflect this more accurate language and references.

f) Supplementary data: It's not clear whether the Supplementary data are a summary (it appears to be) or the entire dataset. I know that the entire dataset is evolving and available, but if what is presented here is a summary then I'd prefer that you deposited a snapshot of the data at the time the analysis was done, to allow attempted replication using the exact same dataset. But I may have got this wrong.

We have added an additional set of data to the Figure.Data worksheet, which includes the snapshot of the raw RRID data. We have also added text to the figure legend for the Supplementary data stating that the included data snapshot is rather out of date and that far more data is available via SciCrunch and CrossRef, which are regularly updated.

g) Although beyond the scope of this study, the use of RRIDs does not verify that a cell line used for a study is indeed the claimed cell line. That would require each lab to authenticate the samples of the cell lines they are using by STR or SNP genotyping. If journals and funding agencies would enforce this, it may help to prevent misidentified cell lines from being unknowingly used in biomedical research. Please consider adding a sentence or two on this topic.

SNP profiling is critical for authenticating cell lines in the literature, we have added text (subsection “Example sentences containing cell lines detected by the SciScore algorithm”) to strengthen our position on this point. However, as the reviewer rightly points out, the presence or absence of genetic profiling of cell lines is not currently in scope for RRIDs or this paper. One heartening point may be that we have begun to train the SciScore algorithm to detect STR profiling sentences. In the future, we would expect that the tool would be able to provide to reviewers a note about STR profiling presence in any paper where cell lines are used. If it works and is used, the tool would act as a check for not only cell line identity, but also authenticity.